# Numerical Modelling of the Thermoforming Behaviour of Thermoplastic Honeycomb Composite Sandwich Laminates

**DOI:** 10.3390/polym16050594

**Published:** 2024-02-21

**Authors:** Varun Kumar Minupala, Matthias Zscheyge, Thomas Glaesser, Maik Feldmann, Holm Altenbach

**Affiliations:** 1Fraunhofer Institute for Microstructure of Materials and Systems IMWS, 06120 Halle, Germany; matthias.zscheyge@imws.fraunhofer.de (M.Z.); thomas.glaesser@imws.fraunhofer.de (T.G.); maik.feldmann@imws.fraunhofer.de (M.F.); 2Polymer Processing, Merseburg University of Applied Sciences, 06217 Merseburg, Germany; 3Institute of Mechanics, Otto-von-Guericke University Magdeburg, 39106 Magdeburg, Germany

**Keywords:** composite sandwich, thermoforming, thermo-mechanical analysis, finite element modelling

## Abstract

Lightweight component design is effectively achievable through sandwich structures; many past research studies in the aerospace and racing sectors (since the 1920s) have proven it. To extend their application into the automotive and other transport industries, manufacturing cycle times must be reduced. This can be achieved by sandwich materials made of continuous fibre-reinforced thermoplastic (CFRTP) cover layers and thermoplastic honeycomb cores. To widen the application of flat thermoplastic-based sandwich panels into complex parts, a novel forming technology was developed by the Fraunhofer Institute of Microstructure of Materials and Systems (IMWS). Manufacturing defects like wrinkling and surface waviness should be minimised to achieve high reproducibility of the sandwich components. Studying different manufacturing parameters and their influence on the final part is complex and challenging to analyse through experiments, as it is time-consuming. Therefore, a finite element (FE) modelling approach is implemented to reduce such efforts. Initially, a thermoforming model is developed and validated with experimental results to check its reliability. Further, different simulations are performed to optimise the novel sandwich-forming process. In this study, a thermoplastic sandwich made of polypropylene (PP) honeycomb core and polypropylene glass fibre (PP/GF) cross-ply as cover layers was used, and its numerical model was executed in LS-DYNA software release R11.2.1.

## 1. Introduction

Thermoplastic composite sandwich structure made of honeycomb core and laminated with unidirectional tape (UD-tape) cross-ply cover layers on the top and bottom sides offers reduced mass and high bending stiffness. The honeycomb structure comprises an array of thin-walled, hollow hexagonal cells arranged together, resulting in minimal material per unit area. In addition to that, the honeycomb structure provides relatively high out-of-plane compression and out-of-plane shear properties. Meanwhile, cross-ply cover layers offer a high tensile strength in the fibre direction, resulting in-plane orthotropic properties.

The production of hexagonal honeycombs at an industrial scale is carried out through tube extrusion, expansion, and corrugated processes [1,2,3]. These processes are operated at multiple stages and require long production times, which in total results in high production costs and therefore limits them to a low-volume production [4]. To solve this, ThermHex Waben GmbH has introduced a continuous production process where a thin thermoplastic polymer film is extruded (1st Step) and vacuum formed to form a half hexagon (2nd Step), followed by folding to form a honeycomb (3rd Step), and at the end lamination with cover layers (4th Step), resulting in a composite sandwich structure as shown in Figure 1a. The so-produced semi-finished laminate is called Organosandwich, which is aimed at mass production with maximum production speeds of up to 20 m/min and a panel width of 1.2 m [5].

With the added advantage of thermoplastic material behaviour and suitability for high volume production, the flat semi-finished panels can be further processed into complex shapes using the thermoforming technique so that they can be used in different applications, especially in the automotive sector. To form and include functionalizing elements into sandwich components, a multi-stage thermoforming technique known as Thermoplastic Sandwich Moulding (TSM) is developed at Fraunhofer IMWS, with a manufacturing cycle time of 4 s [6]. Simultaneously, with the process technique approach, Fraunhofer IMWS develops the numerical models and performs simulations to adapt the novel technology to the digital process chain.

## 2. Thermoplastic Sandwich Forming

The TSM technology allows the processing of semi-finished thermoplastic sandwich panels into ready-to-use lightweight parts. The focus of this research is to develop a manufacturing process that allows the honeycomb core sandwich laminate to conform to the mould shape without wrinkling or the core collapsing or buckling. The novel TS-moulding manufacturing process of FRTP sandwich components comprises three main process steps, as shown in Figure 1b, starting with IR-heating of the semi-finished sandwich laminate, followed by forming the sandwich laminate into a pre-defined shape, and then injection moulding around the edges [6].

Specifically, thermoforming is carried out through a press by closing the two-sided tool in three successive sub-process steps. In the first step of the closing movement, the sandwich structure is formed in areas of the later part in which the core must remain intact. After this forming step, the press stops the closing movement for a short time, allowing cooling by tool contact and thereby solidifying the face sheet matrix. After this short break, the tool moves further and fully closes by pressing the sandwich edges into a compact laminate. Pressing is particularly important at the component edge for sealing the sandwich to protect the core and later for the integration of functional elements. The pressed edge also acts as a transition zone between the formed sandwich structure and over-moulded edges [4].

## 3. Materials and Finite Element Modelling

Using mathematical models, physics laws, and computer science, it is essential to reduce the physical work and save the extensive time that is required to conduct numerous experiments through a trial-and-error approach. The finite element analysis (FEA) approach starts with material description and material characterization, followed by FE-modelling with necessary boundary conditions and loading cases. Simulation results are used to identify undesirable deformations, in part at the early stages of thermoforming process development, as shown in Figure 2.

### 3.1. Sandwich Material Description

The Organosandwich used for this study comprised a PP honeycomb core and PP/GF UD-Tape-based Cross-Ply cover layers, the sandwich specifications are provided in Table 1. During the production of such laminates, a thin PP film is formed into a honeycomb core and laminated with the PP/GF cross-ply using heating and a double-band press. This lamination allows a strong weld joint between the core and cover layers as they share the same thermoplastic material. There is no requirement for additional adhesive layers, unlike in Nomex or another type of sandwich laminate, making the production more sustainable and economical.

### 3.2. Cover Layers Wrinkle Analysis

The thermal and mechanical loads on sandwich components result in different deformation modes, on the one hand, cover layers exhibit following behaviour:ply elongation;inter-ply slip;ply shear; andwrinkle or in-plane buckling [8,9].

On the other hand, the core component exhibits in-plane tension and compression and out-of-plane buckling due to compression [2]. In thermoforming process development, the focus is to produce a part without any defects. Sometimes, CFRTPs such as woven laminate or non-woven laminate (cross-ply) during thermoforming show surface defects, which usually develop due to buckling or wrinkling. The wrinkle formation in cover layers of non-homogeneous sandwich materials is treated as a local phenomenon. This kind of deformation can be initially estimated through a model of axially loaded flat sandwich structure (see Figure 3). The prediction of onset wrinkle in cover layers is performed with the calculation of the maximum compressive stresses in the cover layer.

There exist different mathematical models to estimate the maximum compressive stress for the initiation of wrinkle formation in the cover layers, based on the homogeneous and non-homogeneous core materials. For non-homogeneous cores, isotropic cover layers, and where the height of the core is relatively high compared to the thickness of the face sheet (see Figure 1a), Williams [10] made a suitable analysis. This analysis is based on the initial assumption that the axial stress in the direction of the load on the core is zero, known as the anti-plane stress assumption [9]. The core here is infinitely thick; therefore, wrinkles are exponentially decaying on the cover layers away from the core axis. The mathematical expression of wrinkle formation stress for cover layers is derived as
(1)$$\sigma_{wr}=\left[\frac{0.825}{{\left(1-{\nu_{cl}}^{2}\right)}^{1/3}}\right]{\left(E_{cl}E_{c}G_{c}\right)}^{1/3},$$
where $\sigma_{wr}$ is the maximum compressive stress for wrinkle formation, $\nu_{cl}$ is the Poisson’s ration of the cover layer, $E_{cl}$, $E_{c}$ are the Young’s modulus of cover layer and core, respectively, $G_{c}$ is the in-plane shear modulus of the core. An additional assumption is made in the current study, based on the thermoplastic sandwich lamination, where additional molten polymer is deposited in the core and cover layer bonding regions. Therefore, orthotropic cover layers are considered locally isotropic in the contact region between core and cover layers, calculating Equation (1) for Organosandwich at processing temperature, the maximum in-plane compressive stress which result in wrinkle formation is estimated as below:(2)$$\sigma_{wr}=\left[\frac{0.825}{{\left(1-0.4^{2}\right)}^{1/3}}\right]{\left(16.27*130*0.09\right)}^{1/3}=5.03\;N/{mm}^{2}.$$

### 3.3. Spatial Discretisation of Organosandwich

TS-moulding is a combined process of thermoforming and over-moulding; however, only thermoforming is discussed in this paper. The demonstrator part used for this study comprises different zones, like forming and pressing regions. Therefore, the sandwich is divided into three parts: a honeycomb core, top and bottom face sheets, where each part is sub-modelled using shell element theory. The honeycomb structure is generally modelled using three methods: (1) the meso model, (2) the volume model, and (3) the shell model. In relation to the manufacturing process, where the honeycomb structure significantly influences the formation locally and globally, the meso model approach is chosen in this study. Using mesh discretization, a complex geometry is divided into smaller entities and simpler subdomains. Then, their respective partial differential equations are solved by applying the principles of the boundary value problem. During the selection of element types and their related interpolation functions, the physical quantity for, e.g., displacement, temperature, etc., is considered.

In the finite element method approach, the strong form of the initial boundary value problem for small displacements is written as follows:(3)$$\nabla \cdot \sigma +b=\rho \ddot{u}\;in\;Ω,$$ where σ is stress, b is body force, ρ is density, and $\ddot{u}$ is the time derivative of displacement, with the auxiliary conditions like, essential boundary conditions $u_{i}={\bar{u}}_{i}\;on\;\Gamma_{u_{i}}$ and natural boundary conditions $\sigma_{ij}n_{j}={\bar{t}}_{i}\;on\;\Gamma_{f_{i}}$. The weak form is then written as:(4)$$\int_{\Omega }^{\;}\rho \;\delta u\cdot \ddot{u}d\Omega +\int_{\Omega }^{\;}\delta \varepsilon ∶\sigma d\Omega =\sum_{i=1}^{n}\int_{\Gamma_{f}}^{\;}{\delta u}_{i}{\bar{f}}_{i}d\Gamma +\int_{\Omega }^{\;}{\delta u}_{\;}\cdot b\;d\Omega ,\;$$
which is used to develop the discrete equations in the Galerkin procedure [11]. The displacement approximation of standard FEM method is defined through shape functions, for a displacement $u_{i}$ of an isoparametric element, the interpolation function is,
(5)$$u_{i}^{h}\left(X,t\right)=\sum_{I=1}^{n_{ne}}N_{I}\left(\xi \left(X\right)\right)u_{ij}\left(t\right),X\in Ω,$$
where $n_{e}$ is the number of nodes of elements, for a standard four node element the shape functions $N_{I}$ are written as:(6)$$N_{I}\left(\xi \right)=\frac{1}{4}\left(1+\xi_{I}\xi \right)\left(1+\eta_{I}\eta \right),$$
here ξ(X), η(X) are the expressions of mapping of physical domain to its parent domain [11].

### 3.4. Thermo-Mechanical Modelling

Thermoforming is a combined problem of mechanical analysis and thermal analysis, where stress generation due to mechanical deformation as well as temperature variation are vital. In the thermo-mechanically coupled finite element model, accurate evaluation of mechanical stresses and thermal stresses at each point of the physical subdomain must be conducted under the influence of temperature. The heat conduction in the structure is governed by Fourier’s law of conduction [12]. Straining in the structure generates heat in transient and dynamic processes [13]. Thermo-mechanical material model assumed here a three-dimensional non-linear, transient heat conduction problem for a domain Ω bounded by Γ, the governing differential equation written as:(7)$$\frac{\partial }{\partial x}\left(k_{x}\frac{\partial T}{\partial x}\right)+\frac{\partial }{\partial y}\left(k_{y}\frac{\partial T}{\partial y}\right)+\frac{\partial }{\partial z}\left(k_{z}\frac{\partial T}{\partial z}\right)+Q\left(x,y,z,t\right)=\rho c\frac{\partial T\left(x,y,z,t\right)}{\partial t}.$$
where


$T\left(x,y,z\right)$ is the temperature at point and for time t;$Q\left(x,y,z,t\right)$ is the internal strength of heat source;ρ is the density;c is the specific heat;$k_{x},k_{y},k_{z}$ are thermal conductivities in the x,y,z directions, respectively, at initial conditions t=0, temperature is $T_{0}$ [14].


The standard mesh FE model has an inherent, overly stiff phenomenon that causes poor accuracy in solving complex problems. To address this Liu, ref. [15] applied the smoothing gradient technique and established the smoothed Galerkin weak form. Further, a node-based smoothing finite element method was introduced to construct the smoothing domain of transient thermal mechanical analysis to obtain better results than standard FEM [14]. The balance equations for the coupled problem are composed of a force field and a thermal field balance equation [16]. By substituting the thermal shape functions of element nodes in the Galerkin weak form for transient heat conduction, the discretised system of equilibrium equations is expressed as the following matrix form:(8)$$N_{T}\left\{\frac{\partial T}{\partial t}\right\}+\left[K_{T}+K_{T}^{h}\right]\left\{T\right\}=\left\{P\right\},$$
where $N_{T}$ is the thermal capacitance matrix, $K_{T}$ is the conduction matrix, P is the thermal load vector of the predefined thermal shape functions. After defining the initial temperature field, the respective values at an arbitrary time t can be approximated by applying the backward difference technique to Equation (8).

The governing equations for transient thermo-mechanical problem over the specific domain Ω bounded by Γ are written as,
(9)$$\begin{array}{c} \mathrm{Equilibrium}\;\mathrm{equation}:\;\sigma_{ij,j}+b_{i}=0,\; \end{array}$$
(10)$$\mathrm{Displacement}\;\mathrm{boundary}:\;u_{i}=u_{\Gamma }\;\mathrm{on}\;\Gamma_{u},$$
(11)$$\mathrm{Stress}\;\mathrm{boundary}:\;\sigma_{ij}n_{i}=f_{\Gamma }\;\mathrm{on}\;\Gamma_{f},$$
(12)$$\mathrm{Strain}\;\mathrm{displacement}\;\mathrm{relations}:\;\varepsilon_{ij}=\frac{1}{2}\left(u_{i,j}+u_{j,i}\right),$$
where $\sigma_{\;}$, ε and u are the stress, strain, and displacement, b is the body force, $u_{\Gamma }$ and $f_{\Gamma }$ are the displacement and traction on the essential and natural boundaries. The above stress and strain are an expression of the thermal expansion coefficient,
(13)$$\sigma_{ij}=\delta_{ij}\lambda \varepsilon_{kk}+2\mu \varepsilon_{ij}-\delta_{ij}\left(3\lambda +2\mu \right)\alpha ∆T\left(t\right),$$
in the above equation λ and μ are Lame’s constants, they can be derived from the Poisson’s ratio and Young’s modulus, $\delta_{ij}$ is the Kronecker delta, α is the thermal expansion coefficient, and $∆T\left(t\right)$ is the change in temperature with the time t. In Equation (10), $\varepsilon_{ij}$ is the total strain, which is sum of elastic, plastic, and thermal strain,
(14)$$\varepsilon_{ij}=\varepsilon_{ij}^{e}+\varepsilon_{ij}^{p}+\varepsilon_{ij}^{T}.$$

With the temperature field obtained, the thermal stress and strain analysis can be performed using the standard Galerkin weak form,
(15)$$\int_{\Omega }^{\;}{\left\{\delta \varepsilon (u)\right\}}^{T}\left\{\sigma \right\}d\Omega -\int_{\Omega }^{\;}{{\left\{\delta u\right\}}^{T}\left\{b\right\}}_{\;}d\Omega -\int_{\Gamma_{f}}^{\;}{\left\{\delta u\right\}}^{T}\left\{f_{\Gamma }\right\}d\Gamma =0,$$
where u is the displacement field, which is interpolated using linear shape functions as below,
(16)$$u=\sum_{i=1}^{n}\Phi_{i}d_{i},$$
$d_{i}$ is the nodal displacement of the node i, and $\Phi_{i}$ is the linear shape function. By solving Equation (15) using Equation (16), the discretised system of equations for a thermo-mechanical problem can be expressed in the form of matrix as follow,
(17)$$K_{d}\left\{d\right\}=\left\{F\right\},$$
where $K_{d}$ is the stiffness matrix comprised of product of elastic coefficient matrix and strain displacement matrix, F is the load vector. For time-dependent non-linearity of materials Equation (11) is solved by using Newton–Raphson iteration method at each time step.

### 3.5. Materials and Methods

To develop a thermo-mechanical FE model, the thermo-mechanical properties and thermo-physical properties of sandwich components should be measured accordingly. For thermo-mechanical properties, like temperature-dependent E-modulus E (T), and Stress–strain curves σ−ε (T), a standard uniaxial tensile test is conducted at elevated temperatures according to the norm DIN EN ISO 527-2 (test specimen type 1A) [17], with a traverse speed of 50 mm/min. The tensile tests are performed in a closed environment of temperature chamber, see Figure 4a, an average of five experiments for each temperature is plotted in Figure 4b.

The existing material model in LS-DYNA, MAT_Elastic-Plastic_Thermal, which uses uniaxial tensile test data from Figure 4b as input, is chosen for the honeycomb core. Cross-ply is calibrated according to the fibre-matrix weight composition of 60% (PP/GF60) on a single UD-tape with MAT_Reinforced_Thermoplastic, in which matrix properties and fibre properties are defined separately. The calibrated materials for PP and PP/GF60 cross-ply according to the single element test simulation are shown in Figure 5a and Figure 5b, respectively.

The thermo-physical properties like temperature-dependent thermal conductivity k (T) and specific heat capacity c (T) of both core and cover layers are measured through laser flash analysis (LFA) and differential scanning calorimetry (DSC) experiments, respectively. The experimentally obtained data for PP and PP/GF60 are presented in Figure 6a,b. The thermo-physical experimental data are used as input in the MAT_Thermal_Isotropic for the core. Assuming that heat transfer in UD-tape is the same along the fibre direction and transverse direction $k_{x},k_{y},k_{z}$ = 1, MAT_Thermal_Isotropic is chosen for each laminate in the cross-ply.

### 3.6. Boundary Conditions

The thermoforming model developed here is aimed at describing all three geometric, material, and boundary non-linearities of the thermoforming process of sandwich laminate. For this, a simplified geometry model is developed, considering the actual geometry. A unit-cell representation of the sandwich and honeycomb core is shown in Figure 7a. Hence, L is the length orientation, W is the width orientation, and H is the sandwich thickness and out-of-plane direction. The shell element formulation for cell walls as well as the face sheet is conducted using the Reissner–Mindlin kinematic assumption.

The thermal state of the sandwich laminate as shown in Figure 7b is achieved through the IR-heating process, where the sandwich laminate is heated from the top and bottom sides so there is a symmetrical thermal state through the thickness. Numerous experiments were conducted by controlling the intensity of the heat and heating time. Glaesser explained this technique in detail in his thesis work [6]. The finalised temperature profile of the sandwich, where the face sheet is brought to matrix melt temperature T_m_ and the core is at varying temperatures from T_g_ and below, as shown in Figure 5b.

Contact between the core and cover layers plays a significant role in thermoforming. Pre-heating of sandwich laminate in the thermoforming process (see Figure 1b), at which cover layers reach melting point, results in sandwich disbonding. The contact interface in melted thermoplastics is viscosity-dependent, showing static and dynamic friction regimes. This type of contact is modelled using a Tie-Break contact, and the governing equation for such contact is expressed as:(18)$$\mu_{\;}=\mu_{D}+(\mu_{S}-\mu_{D})e^{-DC(\vartheta )}$$
where $\mu_{\;}$ is the co-efficient of friction, suffix S and D represent static and dynamic co-efficients, respectively, DC is the decay constant between static and dynamic regimes.

As shown in Figure 8a, in the contact interface, the nodes of a honeycomb core are tied to the cover layer initially. During thermoforming, the contact interface is governed by the friction co-efficient, as in Equation (18). An approximated contact behaviour at thermoforming temperature is presented in Figure 8b.

## 4. Results and Discussion

### 4.1. Thermoforming

Thermoforming results of the PP and PP/GF-based Organosandwich components are shown in Figure 9a, which is a simplified sandwich model formed between positive and negative tool forms. The form structure is a demonstrator part that is developed to analyse different forming mechanisms and forming parameters.

Here, we show the optimised thermoforming results with controlled laminate temperature and forming velocity. Owing to the sandwich bending principle, the top face sheet experiences in-plane tension, while on the other side, the bottom face sheet experiences in-plane compression [9]. With tension and compression on the top and bottom sides, the core exhibits pure out-of-plane shear deformation under bending [18]. The cross-ply as face sheet, which has fibre orientations at 0° and 90° in the Y and X directions, respectively, shows in-plane shear stresses in the 3D forming area [18], as shown in Figure 9b. In the 3D forming region, as shown in Figure 9b, it is observed that shear stresses are highly concentrated and larger in magnitude compared to the 2D forming region. The in-plane shear stresses will develop axial compression in the transition between shear compression and shear tension. Therefore, unwanted deformations like wrinkle formation, as shown in Figure 10, occur only in distinctive areas of the cross-ply laminate, depending on the fibre orientation [9].

### 4.2. Critical Stress Analysis

For a sandwich demonstrator part, the thermoforming results show a surface waviness on the bottom cover layer in the 3D forming region, as shown in Figure 10 from experiments and Figure 11 from simulations. Sandwich bending accounts for global tension on the top side and exerts global compression on the bottom side. Examining a cross-ply under forming or draping, whilst fibres in the principal axis are subjected to tension, the transverse fibres show compression behaviour. Combining these two effects, the in-plane instabilities likely to happen on the bottom face sheet are wrinkling or buckling. With the detailed study of the results and stress plots in the specific region, the outer ply oriented at 90° on the bottom face sheet is subjected to local tension in the X-direction, and the inner ply oriented at 0° on the bottom face sheet is subjected to local compression in the Y-direction.

In Figure 11, the inner UD-ply oriented in 0° to the *Y*-axis shows a maximum tensile stress of 17 MPa and a maximum compressive stress of 5.2 MPa. The axial compressive stresses in the respective ply are above the maximum critical compressive stress for wrinkle formation, as mentioned in Equation (2), $\sigma_{wr}$ which is 5.03 MPa. As the maximum compressive stresses in the ply exceed the limit, wrinkle formation on the cover layer is inevitable. The prediction of the mathematical model and simulation results show critical regions in the thermoforming of Organosandwich, which can be used for design or process optimisation.

### 4.3. Considerations and Product Optimisation

From thermoforming simulations, the identified critical regions can be rectified with different approaches like optimisation of process parameters, boundary conditions and design modification. Although process parameters like laminate temperature and tool velocity influence both in-plane and out-of-plane buckling, the wrinkle effect in Figure 10 is not their influence. From numerous experiments, the thermal state of the laminate and velocity ramp of the tool are optimally finalised; further modifications in these parameters lead to adverse effects like buckling. The second approach is modification of boundary conditions, which means the holding position of the laminate inside the tool with grippers and clamps. There is already a clamping system in the tool around the U-profile (which acts like a clamping ring in classical forming technologies [19]), which minimises the bending stresses; its influence was discussed in the previous work [20]. In composite laminate forming, this type of wrinkle is rectified by using a spring-loaded frame gripper system; the springs act as tensioners and generate in-plane tension during forming in the respective direction [21]. The pre-consolidated laminate is placed inside a frame with specific springs and transported to the heating station and forming machine. This kind of gripper system inside the process slows down the manufacturing cycle as hitching and removing the laminate will add additional steps. With time constraints, the idea of including additional steps in the process is not taken into consideration. The existing vacuum gripper system inside the tool in the middle of the form cavity stays as the standard [22].

The direct basis of optimisation is to remove critical axial compressive stresses. When existing process parameters do not allow further modifications, the remaining option is to design modifications. To make any changes to the existing part design, one needs to investigate the forming mechanisms at first and their influence on the structural performance subsequently. As the stress concentration is limited to the 3D region, a minor change is made to the existing shell structure, as shown in Figure 12.

The existing design, as shown in Figure 12a, of uniform thickness through the structure is modified to a reduced thickness of approximately 10% in the three-dimensional bending region, as shown in Figure 12b. This reduced thickness generates out-of-plane compression inside the sandwich, and as a reaction, an in-plane local tension is developed, which counters the previously existing critical wrinkle stress. The results of the modified part design are shown in Figure 13. The maximum compressive stresses in the Y-direction for UD-ply orientation at 0° are 3.5 MPa, whereas the average maximum compressive stresses are around 1 MPa, see Figure 13a, which are way below the limit of $\sigma_{wr}$ 5.20 MPa. With this design modification, there is a drop in maximum axial compressive stresses by 33% during forming. Cross-ply of 0°/90° orientation is good at tensile loads on the sandwich structure. Therefore, the top cover layer on the demonstrator part shows no defects in both designs.

Thermoforming trials with a modified tool design in the 3D bending region, as shown in Figure 12, resulted in a wrinkle-free surface [23,24]. Likewise, the simulation results satisfy the experimental trials, as both Figure 13a,b show improved surface on the bottom face sheet when compared to Figure 11 and Figure 10, respectively. The only adverse effect of this optimisation could be the thickness reduction of the formed sandwich part in the specific region.

## 5. Conclusions

The presented study examines the material behaviour of thermoplastic sandwich components under thermo-mechanical loads. A finite element model is developed considering all three: material, geometry, and boundary non-linearities. The initial experimental trials and simulation results identified critical areas on the thermoformed sandwich demonstrator part [25]. Considering the process limitations, as mentioned in Section 4.3, a design modification was performed on the tool geometry, which resulted in reduced stresses on the critical areas of the thermoformed part both in simulations and experimental trials.

Wrinkle deformations in a CFRTP part could directly influence its structural properties, decreasing the tensile and compressive strength by at least 20–30% [26]. It is necessary to identify the critical regions in a part that arise from the manufacturing process and, therefore, eliminate them with optimal solutions. The novel TSM technology was aimed at a high-volume production process in automotive applications. Hence, the numerical method developed in this study can be used in the early stages of part design and thermoforming process development.

## Figures and Tables

**Figure 1 polymers-16-00594-f001:**
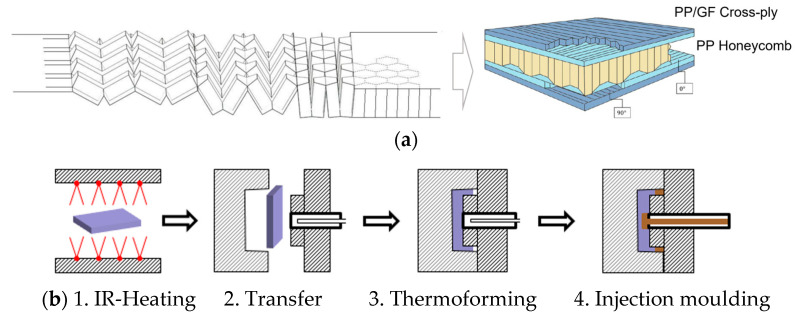
Organosandwich (**a**) continuous production of honeycomb core and in-line lamination of cover layers, (**b**) schematic illustration of the novel manufacturing process of TSM [3,4].

**Figure 2 polymers-16-00594-f002:**
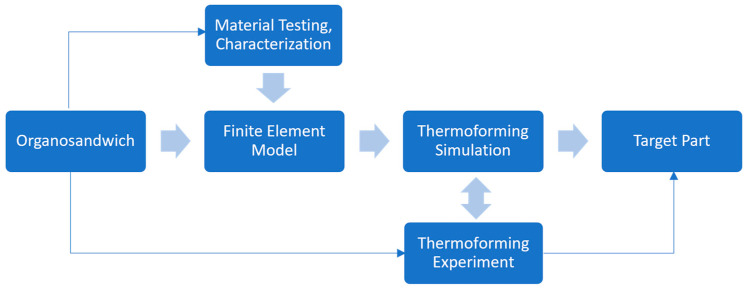
FEA workflow of process simulation of thermoforming and part design.

**Figure 3 polymers-16-00594-f003:**
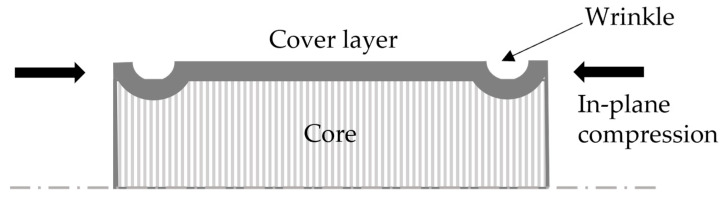
Localised non-harmonic wrinkle mode of sandwich cover layers [9].

**Figure 4 polymers-16-00594-f004:**
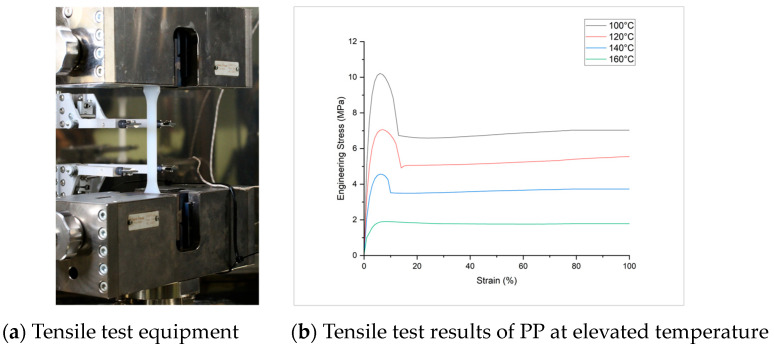
Thermo-mechanical tensile testing of PP at different temperature.

**Figure 5 polymers-16-00594-f005:**
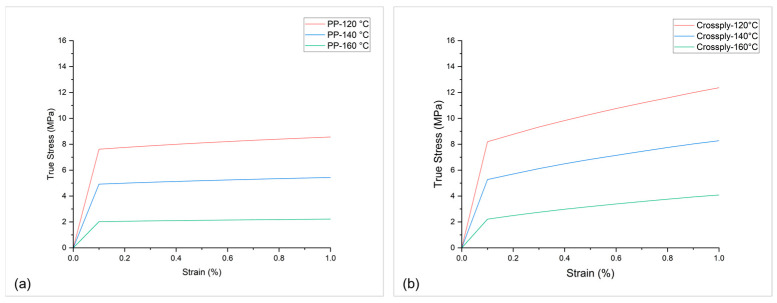
Calibrated material at different temperatures for (**a**) unfilled PP, (**b**) PP/GF cross-ply.

**Figure 6 polymers-16-00594-f006:**
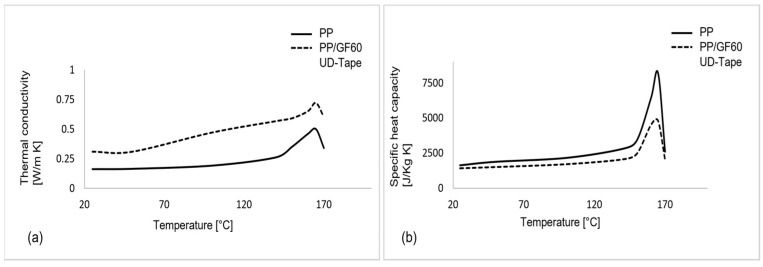
Thermo-physical material properties of core and cover layers: (**a**) temperature dependent thermal conductivity, (**b**) temperature specific heat capacity.

**Figure 7 polymers-16-00594-f007:**
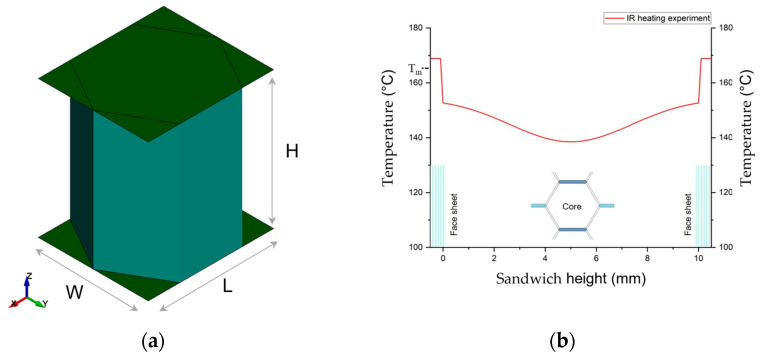
(**a**) Representative idealised sandwich honeycomb cell. (**b**) Initial thermal state of the sandwich after heating step in thermoforming process.

**Figure 8 polymers-16-00594-f008:**
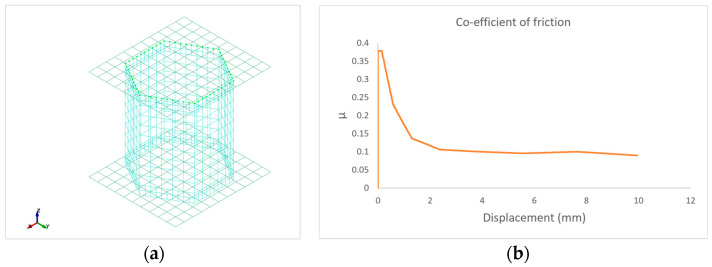
Simulation showing: (**a**) Thermoformed Organosandwich 3D−shell demonstrator, and (**b**) shear stresses in the bottom cover layer.

**Figure 9 polymers-16-00594-f009:**
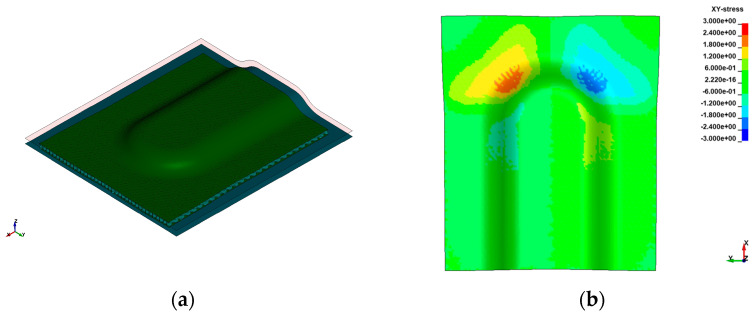
Simulation showing: (**a**) Thermoformed Organosandwich 3D-shell demonstrator, (**b**) shear stresses in the bottom cover layer.

**Figure 10 polymers-16-00594-f010:**
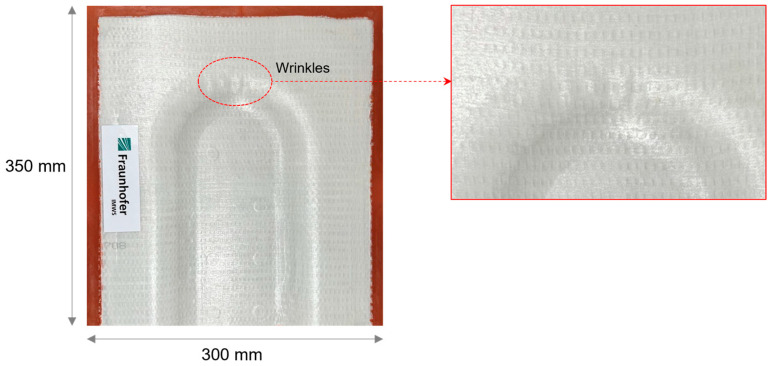
Thermoformed Organosandwich 3D−shell demonstrator part showing wrinkles on bottom cover layer.

**Figure 11 polymers-16-00594-f011:**
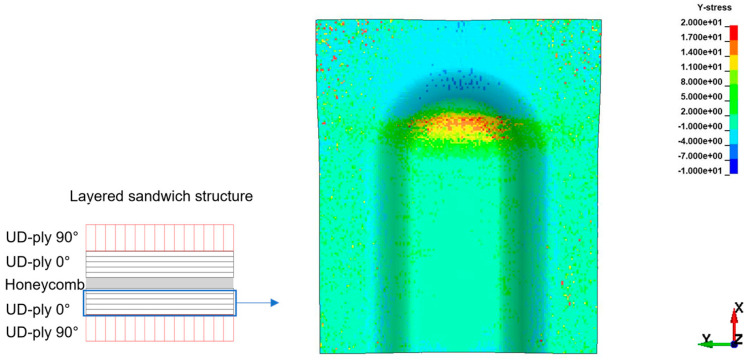
Compressive stresses inside the inner ply of bottom layer, indicating wrinkles [in MPa].

**Figure 12 polymers-16-00594-f012:**
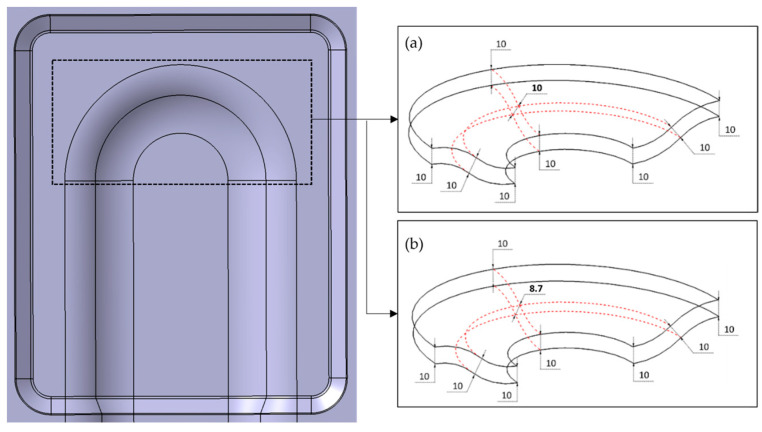
3D–shell demonstrator, (**a**) original design with uniform thickness, (**b**) modified design with reduced thickness [6].

**Figure 13 polymers-16-00594-f013:**
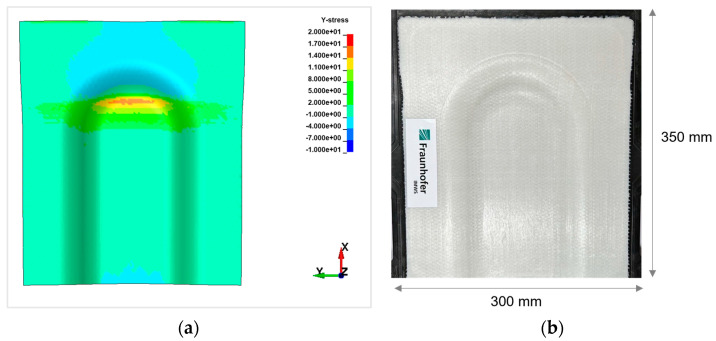
Modified design results (**a**) compressive stresses in the UD-ply of bottom cover layer, (**b**) thermoformed part of modified design.

**Table 1 polymers-16-00594-t001:** Sandwich material description of the used configuration 12THPP120CP820 [7].

Sandwich Entity	Dimension
Sandwich thickness	12 mm
Cover layer thickness	0.5 mm
Honeycomb core thickness	11 mm
Cell size	5 mm
Standard panel dimension	1200 mm × 2500 mm (L × W)
Weight per unit density	3120–3240 g/m^2^

## Data Availability

The relevant data supporting the results presented in this work are published by V.K. Minupala [4] and T. Glaesser [6].

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
