# Peer review of "Numerical Modelling of the Thermoforming Behaviour of Thermoplastic Honeycomb Composite Sandwich Laminates"

_polymers, 2024, doi:10.3390/polym16050594_

Round 1

Reviewer 1 Report

Comments and Suggestions for Authors

1.For the drawing in Figure 1, it is preferable to place the text below (b) for better comprehension.

2.Some small figures can benefit from localized zoom-ins to enhance expressiveness, for example, in Figures 1, 3, 5, and 7.

3.Lines 173-175 could be presented in a table format to illustrate the symbols and their corresponding values.

4.Consider incorporating direct comparisons between simulation and experimental results, for instance, in Figure 10.

5.Introduce a discussion section to provide analysis and arguments.

6.In the conclusion, revisit and elaborate on key points for a more specific presentation.

7.The literature review is insufficient. It is suggested to add several references such as: Influence of size effect on the dynamic mechanical properties of OFHC copper at micro-/meso-scales; Ductile fracture behavior in micro-scaled progressive forming of magnesium-lithium alloy sheet ; Digital twin-driven product design, manufacturing and service with big data.

Author Response

First of all thanks for your time and consideration.

  1. For Figure 1, I have changed the text position for better comprehension
  2. Figure were changed with necessary localized zoom-ins
  3. Lines 173 - 175 are written as bullet points now
  4. Direct comparison is added
  5. Discussions were made along with the results, a separate discussion chapter could be a repetition
  6. Conclusion is re-phrased to keep it simple and convey the subject
  7. This study was focused on thermo-mechanical modelling and one of the possible defects likely to happen in sandwich materials. There are other deformation modes like ply slip, core buckling and influence of cell size and ply-orientation that will be presented in the next publication.

Reviewer 2 Report

Comments and Suggestions for Authors

Reviewer comments 

Manuscript ID: Polymers-2812177

Title: Numerical modelling of the thermoforming behavior of thermoplastic honeycomb composite sandwich laminates

Journal: Polymers

The presented study examines the material behavior of various sandwich components under thermo-mechanical loads. Additionally, a numerical model has been presented to study the thermoforming process of thermoplastic honeycomb composite sandwich laminates.

The paper falls within the scope of the journal; however, I believe it requires some enhancement. I propose the following revisions to improve the paper.

1- Avoid unnecessary details in the abstract and rewrite it; keep it concise and focused. Exclude background information that does not directly contribute to the understanding of your research. Ensure readers can clearly identify the essential components of your paper in the abstract, including the purpose, methods, results, and conclusions.

2- Abbreviations and symbols should be identified the first time they appear, see line 32 (UD-tape). 

3- The novelty of this article should be emphasized and should be addressed clearly in the introduction regarding what you have developed, found, or improved compared with the other similar research’s published before.

4- In Eq. 3: b parameter name should mentioned.

5- Specimen geometry used for tensile tests should be presented in the manuscript

6- Elastic-visco-plastic stress-strain curves for different temperature should be presented

7- Equation for thermo-mechanical behavior incorporated in the software should be added in the manuscript.

8- How the authors adjusted mesh size to the geometry of presented model? What mesh refinement algorithm was used in order to obtain optimal mesh size for presented analyses? If mesh refinement was performed manually, appropriate Figure presenting i.e. stress-mesh size dependency should be included in the manuscript.

9- More details regarding the boundary conditions of the Organosandwich components during the thermoforming operation should be added.

10-  Line 338: behavior not bahavior

11- Typos and grammar problems need to be corrected properly, authors should carefully check through the manuscript before submitting a revision.

The reviewer recommends that the author do major revision to the manuscript. 

Comments on the Quality of English Language

The reviewer would recommend that the authors proofread the article thoroughly for typos and grammatical errors.

Author Response

Thanks for your valuable time and important suggestion, the corrections are taken into count as following,

  1. Abstract is rewritten
  2. Abbrevation for UD (Uni-directional) was mentioned
  3. I tried my best to emphasize the work, as the work majorly belongs my PhD work I was restrciting myself to not publish confidential data (I have simplified it)
  4. In Equation 3, b is body force (mentioned it now)
  5. Specimen geometry I have mentioned in the text and norm, as there were already enough images in the paper I couldn't add
  6. The core is a Elastic-Plastic Thermal material. Elastic-Visco-Plastic Thermal material was used neglecting viscous parameters, from literature research I have found thermal influence is greater on the E and Sigma than the viscous parameters. Only in this material we can import tensile test data as curves, therefore purposefully this material card was used ignoring Cowper and Symonds parameters.
  7. Equation 13 is the thermo-mechanical material model incorporated in the software
  8. The mesh size was adjusted based on the honeycomb cell wall, mesh adaptivity was used in the contact regions and coarse mesh is used in the middle of the core. Mesh study was published in a Master-Thesis already.
  9. The only important missing boundary condition was contact between core and face sheet, which I have added now
  10. I have changed it
  11. Other typos and grammar mistakes were identified and modified accordingly.

This article was submitted to showcase the thermoforming model development and its reliability compared to the experimental trials. Although this model can not trace all realistic deformation modes, it shows evidential results for process optimization. Different deformation modes in the sandwich both in core and face sheets will be explained in upcoming publications.

I sincerely thank you for your inputs, hope the paper looks now in a good shape. Any further corrections are likely to be considered.

Round 2

Reviewer 2 Report

Comments and Suggestions for Authors

The authors have provided correct and comprehensive responses to all the questions raised by the reviewer, underscoring their dedication to enhancing the manuscript. In light of the revision carried out, I recommend that the manuscript be accepted for publication.

Comments on the Quality of English Language

ok